# Inhibition of AXL and VEGF-A Has Improved Therapeutic Efficacy in Uterine Serous Cancer

**DOI:** 10.3390/cancers13235877

**Published:** 2021-11-23

**Authors:** Michael D. Toboni, Elena Lomonosova, Shaina F. Bruce, Jo’an I. Tankou, Mary M. Mullen, Angela Schab, Alyssa Oplt, Hollie Noia, Danny Wilke, Lindsay M. Kuroki, Andrea R. Hagemann, Carolyn K. McCourt, Premal H. Thaker, Matthew A. Powell, Dineo Khabele, David G. Mutch, Katherine C. Fuh

**Affiliations:** 1Barnes Jewish Hospital, Department of Obstetrics and Gynecology, Division of Gynecologic Oncology, Washington University, St. Louis, MO 63110, USA; MDToboni@wustl.edu (M.D.T.); lomonosova.elena@wustl.edu (E.L.); brucesf@wustl.edu (S.F.B.); joantankou@wustl.edu (J.I.T.); marymullen@wustl.edu (M.M.M.); aschab@wustl.edu (A.S.); aoplt@wustl.edu (A.O.); hbeck22@wustl.edu (H.N.); dwilke@wustl.edu (D.W.); kurokil@wustl.edu (L.M.K.); hagemanna@wustl.edu (A.R.H.); cmccourt@wustl.edu (C.K.M.); thakerp@wustl.edu (P.H.T.); mpowell@wustl.edu (M.A.P.); khabeled@wustl.edu (D.K.); mutchd@wustl.edu (D.G.M.); 2Center for Reproductive Health Sciences, Division of Biology and Biomedical Sciences, Washington University, St. Louis, MO 63110, USA

**Keywords:** endometrial cancer, uterine serous cancer, bevacizumab, recurrent endometrial cancer, AXL inhibition, AVB-500, angiogenesis, VEGF inhibition

## Abstract

**Simple Summary:**

Uterine serous cancer (USC) is an aggressive form of endometrial cancer. USC constitutes less than 10% of endometrial cancer cases but is responsible for up to 40% of endometrial cancer deaths. Response to standard platinum and taxane therapy is modest, particularly in the recurrent setting. Thus, novel therapeutic strategies are needed to overcome this drug resistance. We hope that our findings provide strong preclinical support for the combination of an AXL inhibitor with bevacizumab in USC clinical trials in the recurrent setting.

**Abstract:**

Endometrial cancer remains the most prevalent gynecologic cancer with continued rising incidence. A less common form of this cancer is uterine serous cancer, which represents 10% of endometrial cancer cases. However, this is the most aggressive cancer. The objective was to assess whether inhibiting the receptor tyrosine kinase AXL with AVB-500 in combination with bevacizumab would improve response in uterine serous cancer. To prove this, we conducted multiple angiogenesis assays including tube formation assays and angiogenesis invasion assays. In addition, we utilized mouse models with multiple cells lines and subsequently analyzed harvested tissue through immunohistochemistry CD31 staining to assess microvessel density. The combination treatment arms demonstrated decreased angiogenic potential in each assay. In addition, intraperitoneal mouse models demonstrated a significant decrease in tumor burden in two cell lines. The combination of AVB-500 and bevacizumab reduced tumor burden in vivo and reduced morphogenesis and migration in vitro which are vital to the process of angiogenesis.

## 1. Introduction

Endometrial cancer incidence continues to rise annually with an estimated 66,570 cases and 12,940 deaths predicted in the United States in 2021 [1]. The five-year overall survival (OS) for all endometrial cancers is over 80% since 75% of cases present with early stage disease. However, for the cases with extrauterine spread at time of diagnosis, survival significantly decreased [2]. In aggressive subtypes, extrauterine spread at diagnosis has been reported in up to 70% of cases, which has a drastic effect on survival [3,4,5]. One of the most aggressive subtypes is uterine serous cancer (USC). Although USC accounts for only 10% of uterine cancer cases, it is responsible for 39% of uterine deaths making it the most deadly uterine cancer [6].

Traditionally, treatment for USC includes hysterectomy, bilateral salpingo-oophorectomy, and staging including omental biopsy, followed by a platinum and taxane doublet [7,8,9,10,11]. However, estimates have shown standard chemotherapy in advanced USC has a modest effect with a 60% response rate and a five-year survival rate of 41% [12]. In the recurrent setting, the majority of recurrences occur outside of the pelvis, and the survival is significantly lower with a response rate of less than 50% [9,13].

In the last two decades, the molecular profile of USC has been better defined, and this has led to development of targeted agents to treat this aggressive cancer. In 2002, overexpression of human epithelial growth factor receptor 2 (HER2)/neu in USC was first reported [14]. This tyrosine kinase receptor regulates cell growth, proliferation, and differentiation [15,16,17]. In 2018, a phase 2 trial of trastuzumab combined with paclitaxel and carboplatin (PC) in advanced or recurrent patients with HER2/neu expressing USC was published. This trial showed promising results with 17.7 months vs. 9.3 months PFS in patients receiving first-line treatment (*p* = 0.015). However, in the recurrent setting, there was a more modest 2.2 month improvement in PFS and no difference in OS. Although this study led to a targeted treatment for 30% of USC tumors expressing HER2/neu, there are still 70% of patients without HER2/neu expression that are in need of new treatment options [18,19,20].

Bevacizumab is a targeted agent that is used to treat uterine cancers without HER2/neu expression. This monoclonal antibody inhibits VEGF-A and has been tested in multiple endometrial cancer trials [21,22,23] that included 14–25% of patients with uterine serous cancer. In the recently published data from the MITO END-2 trial, a sub-analysis showed the benefits of bevacizumab in non-endometrioid histologies, such as in USC tumors [23]. Given these data, it is possible that bevacizumab could have a greater benefit in aggressive subtypes, such as USC; however, improving response to bevacizumab is still needed.

The receptor tyrosine kinase, AXL, is a critical driver of cell survival, proliferation, adhesion, apoptosis, migration, and invasion [24,25]. It is activated by binding of growth arrest specific gene-6 (GAS6), which is its only known ligand. This binding causes dimerization and initiation of an intracellular signaling cascade [24,26]. This cascade includes the PI3K/AKT pathway downstream, which has long been associated with endometrial cancer and has been reported to be mutated in almost 25% of USC [25,27,28]. Recent work in our lab has demonstrated high AXL expression in 71% of USC tumors, which correlated with poor survival when compared with tumors that had low AXL expression. Additionally, AXL expression is elevated in chemoresistant USC tumors [29,30].

The role of AXL in angiogenesis has been previously described in renal cell cancer and glioblastoma multiforme [31,32]. In the last decade, co-expression of AXL and GAS6 was discovered on vascular cells present in various tumor types [33]. Multiple groups have shown antiangiogenic effects of AXL inhibition both in vitro and in vivo, specifically a synergistic effect was seen in combination with anti-VEGF therapy, such as pazopanib [31,32,34].

One novel method to inhibit AXL is through AVB-500, which is an ultra-high-affinity selective and specific fusion protein that works by sequestering GAS6. A recent Phase Ib clinical trial in platinum-resistant ovary cancer showed this was a safe and tolerable agent [35]. This led to a phase 3 trial, GOG 3059/ENGOT OV66, which is currently open to accrual. Our objective is to determine whether inhibiting receptor tyrosine kinase AXL with AVB-500 in combination with bevacizumab would improve the response to the treatment of USC.

## 2. Results

### 2.1. Genetic Inhibition of AXL Downregulates Signal Intensity of Pro-Angiogenic Factors

To determine whether AXL affects key modulators of angiogenesis, we examined whether genetic or therapeutic inhibition of AXL would alter the expression of pro-angiogenic cytokines in culture media [36]. CM from ARK1 shAXL and ARK1 shSCRM was harvested and subjected to cytokine profiling. The ARK1 shSCRM values for each cytokine were set at 1. Pro-angiogenic cytokine values in ARK1 shAXL CM were divided by their corresponding values in ARK1 shSCRM CM to determine a fold change between each cell line. We found that genetic inhibition of AXL resulted in a significant reduction in several secreted pro-angiogenic factors, including VEGF, PDGF, HGF, and IGF (Figure 1A). To further validate these findings, we inhibited AXL in ARK1 cells by treating them with AVB-500, a highly specific inhibitor in the GAS6/AXL signaling pathway. We found that, similar to genetic knockdown, AXL inhibition with AVB-500 also led to decreased levels of multiple pro-angiogenic factors in CM (Figure 1B).

To confirm AXL’s role in the regulation of VEGF, conditioned media from ARK1 shSCRM and ARK1 shAXL cells were analyzed by ELISA. The VEGF percent concentration is significantly lower in the AXL knockdown group (100% vs. 77.3%, *p* = 0.0021), demonstrating that genetic inactivation of AXL can decrease VEGF expression (Figure 1C).

### 2.2. AXL Inhibition Enhances the Effect of Bevacizumab on Angiogenesis

It has been established that bevacizumab decreases HUVEC invasion in vitro [37]. To better understand the effect of AXL on angiogenesis, we examined whether inhibiting AXL with AVB-500 in combination with bevacizumab could further decrease HUVEC invasion. CM from USC cell line (ARK1) was harvested. The four conditions were control, AVB-500 alone (2 µM), bevacizumab alone (250 µg/mL), and bevacizumab plus AVB-500. After Matrigel matrix was primed with the CM, HUVECs were plated. Inserts were stained after 16 h and showed significantly less percent of cell invasion in the bevacizumab plus AVB-500 group than in the AVB-500 only group (27.6% vs. 44.3%, *p* = 0.0002) and the bevacizumab only group (27.6% vs. 40.3%, *p* = 0.0032). There was no statistical difference between the invasion with AVB-500 alone or bevacizumab alone (44.3% vs. 40.3%, *p* = 0.2608) (Figure 2A,B). These findings indicate that AXL inhibition in addition to inhibition of VEGF-A decreases angiogenesis in vitro.

This inhibitory effect was similarly tested through the tube formation assay (TFA). HUVECs were mixed with CM from the four conditions in 2% FBS media. The cells and CM were plated on the RGF BME and incubated. After 6 h, representative pictures were taken and analyzed. Significant differences were seen across all metrics, which are commonly used in the literature [38,39,40,41]. In the analysis, when comparing the bevacizumab plus AVB-500 group to the AVB-500 only or bevacizumab only groups, the bevacizumab plus AVB-500 group had significantly less branching points (12.56 vs. 36.71, *p* < 0.0001; 12.56 vs. 24.06, *p* = 0.0002), total tubes (59.06 vs. 107.10, *p* < 0.0001; 59.06 vs. 84.94, *p* < 0.0001), percentage of covered area (12.58% vs. 20.15%, *p* < 0.0001; 12.58% vs. 16.83%, *p* = 0.0001), and total tube length (5042 pixels vs. 8789 pixels, *p* < 0.0001; 5042 pixels vs. 7020 pixels, *p* < 0.0001) (Figure 2C–G).

### 2.3. Inhibition of AXL and VEGF-A Shows Relative Decrease in VEGF

We examined whether the differences seen in the steps of angiogenesis were related to VEGF levels. To do this, we performed a VEGF ELISA in order to assess whether there were decreased levels of VEGF in the combination vs. bevacizumab alone group. When comparing the bevacizumab plus AVB-500 group to the bevacizumab only group, there was a significant difference in levels of VEGF (1.45 ± 0.4 vs. 2.24 ± 0.5 pg/mL, *p* = 0.023) (Figure 2H).

### 2.4. Inhibition of AXL and VEGF-A Shows Decrease in pAKT

Next, a possible mechanism by which the combination of bevacizumab and AVB-500 decreases angiogenesis was investigated. One downstream signaling pathway shared by both VEGF-A and AXL that contributes to angiogenesis is the PI3K/AKT pathway [24,42,43]. Western blot analysis of HUVECs showed that a combination of bevacizumab plus AVB-500 reduced pAKT to a greater degree compared to bevacizumab or AVB-500 alone (0.21 vs. 0.49; 0.21 vs. 0.59) (Figure 2I). These results offer a potential mechanism by which co-inhibition of VEGF-A and AXL may decrease angiogenesis.

### 2.5. IHC CD31 Staining of Mouse Tumors Shows Decreased Vessel Density

Histological assessment of harvested tumors allow us to investigate differences between each treatment group. To do this, we assessed microvessel density from the tumors of the treated mice. Tumors from each of the four conditions were processed, placed on slides, and stained with anti-CD31 to determine if the smaller tumor size and volume could be due to decreased vessel density. After the slides were imaged, they were quantified at 200×. When comparing the bevacizumab plus AVB-500 group to the AVB-500 only and bevacizumab only groups, there was a significant decrease in the mean vessel density (28.0 vs. 37.5, *p* = *0*.0283; 28.0 vs. 50.5, *p* = 0.0005). Representative images and the vessel density quantification are seen in Figure 3.

### 2.6. AXL Inhibition in Combination with Inhibition of VEGF-A Significantly Reduces Tumor Burden in Mouse Models

Given our observations that the combination of AVB-500 plus bevacizumab was more effective than either drug alone in reducing both tumor cell VEGF secretion and blood vessel formation in vitro, we next sought to determine their effects on tumor growth in vivo. We intraperitoneally (IP) injected mice with the established USC cell lines ARK1 or ARK4; allowed the tumors to establish; and then treated the mice with AVB-500, bevacizumab, or both drugs (Figure 4A and Figure 5A). In the ARK1 model, mice treated with bevacizumab plus AVB-500 had significantly fewer resectable tumor nodules (over 1 mm) than mice treated with AVB-500 alone or bevacizumab alone (4.7 vs. 8.4, *p* = 0.0245; 4.7 vs. 9.0, *p* < 0.0001, Figure 4B). Additionally, mice treated with both drugs had a smaller tumor mass (58.03 mg vs. 216.50 mg, *p* = 0.0257; 58.03 mg vs. 171.60, *p* = 0.0002, Figure 4C) and a smaller tumor volume (75.1 mm^3^ vs. 396.6 mm^3^, *p* = 0.0467; 75.14 mm^3^ vs. 156.0 mm^3^, *p* = 0.0080, Figure 4D).

In the ARK4 model, mice treated with bevacizumab plus AVB-500 had significantly fewer resectable tumor nodules than mice treated with AVB-500 alone or bevacizumab alone (4.25 vs. 12.40, *p* = 0.0004; 4.24 vs. 7.40, *p* = 0.0009, Figure 5B). Additionally, mice treated with both drugs had fewer nodules under 1 mm (18.25 vs. 66.80, *p* < 0.0001; 18.25 vs. 42.40, *p* = 0.0003, Figure 5C), smaller tumor mass (75.93 mg vs. 424.90 mg, *p* < 0.0001; 75.93 mg vs. 182.70 mg, *p* = 0.0007, Figure 5D), and smaller tumor volume (47.5 mm^3^ vs. 338.2 mm^3^, *p* = 0.0002; 47.50 mm^3^ vs. 176.4 mm^3^, *p* = 0.0009, Figure 5E).

Since mice injected with ARK4 cells developed ascites and diaphragmatic metastases during our pilot experiment, we asked whether AVB-500 plus bevacizumab affected these observations. Mice treated with both drugs were significantly less likely to develop ascites and developed less ascites than those treated with either drug alone (Table 1). Likewise, mice treated with both drugs were significantly less likely to develop diaphragmatic metastases than those treated with either drug alone (Table 1). Together, these data indicate that the combination of AVB-500 plus bevacizumab alone is more effective in reducing endometrial tumor growth in vivo than either drug alone (Appendix A).

## 3. Discussion

Women with USC have poor overall survival, largely due to an increased risk of metastasis at diagnosis and a high recurrence rate with the majority of these recurrences occurring outside of the pelvis [6,9]. Here, we present three lines of evidence that the AXL inhibitor AVB-500 in combination with the VEGF inhibitor, bevacizumab, may be effective in treatment of USC. First, AVB-500 plus bevacizumab was more effective at reducing in vivo tumor growth of two USC tumor models than either drug alone. Second, treatment with AVB-500 plus bevacizumab reduced tumor vessel density in vivo more than either drug alone. Finally, the drug combination decreased VEGF concentration, endothelial cell invasion, and tube formation in vitro.

Bevacizumab is approved for use in recurrent endometrial cancer. Aghajanian and colleagues published a phase 2 study of 52 patients treated with single agent bevacizumab in which the objective response rate was 13.5% with a median response duration of 6 months. USC accounted for 26.9% of the total study population, and the one patient with a complete response had uterine serous cancer [21]. Our data suggest that AVB-500 could enhance the response to bevacizumab.

AXL and VEGF-A share a downstream signaling pathway through PI3K/AKT. Activation of PI3K/AKT leads to endothelial cell survival and uncontrolled tumor growth, and it has been previously reported that AXL activation was required for VEGF-A-dependent endothelial cell migration and tube formation [31,42,44,45]. For example, Li et al. showed that AXL promoted tumorigenesis and metastasis, and that inhibiting AXL decreased endothelial tube formation in breast and lung cancers. Inhibiting AXL caused upregulation of the angiopoietin signaling system, which, in the presence of VEGF, leads to angiogenesis. However, when VEGF is also inhibited, upregulation of angiopoietin factors leads to vessel destabilization and cell death [31]. Evidence of this was seen in Figure 2H,I, where co-inhibition of VEGF-A and AXL lead to a greater decrease in VEGF as well as a decrease in p-AKT expression. Importantly, AXL inhibition contributes to a potentially broader deregulation of angiogenesis as it reduces secretion of several other pro-angiogenic factors from cancer cells. This is described in the schematic in Figure 6.

Additionally, renal cell carcinomas, which are highly vascularized and have a sustained response to antiangiogenic therapies, have been shown to upregulate AXL expression [46,47,48,49]. AXL signaling increased renal cell carcinoma expression of the pro-angiogenic factor S100A10, and an AXL inhibitor combined with a VEGF inhibitor decreased the growth of patient-derived xenografts and vessel density in mice [32]. These findings support those demonstrated by our research.

## 4. Materials/Methods

### 4.1. Cell Lines and Culture Conditions

Human USC cell lines that have been immortalized and established in the literature were used [50,51]. ARK1 cells were provided by Shi-Wen Jiang (Mercer University School of Medicine, Savannah, GA, USA), ARK4 cells were purchased from Dr. Santin (Yale University, New Haven, CT, USA), and human umbilical vein endothelial cells (HUVEC) were ordered from Lonza, Switzerland. ARK1 and ARK4 cells were maintained in RPMI-1640 Medium (Sigma-Aldrich Burlington, MA, USA) supplemented with 10% FBS (Sigma-Aldrich Burlington, MA, USA), and 1% penicillin and streptomycin (Thermo Fisher Scientific Waltham, MA, USA). HUVECs were cultured in endothelial culture medium (CC-3156, Lonza) with added growth medium 2 supplement (C-39211, PromoCell Sickingenstraße Heidelberg, Germany). All cells were maintained at 37 °C in a 5% CO_2_ incubator. All cell lines were confirmed negative for mycoplasma prior to any assay by the MycoAlert Mycoplasma Detection Kit (Lonza Gampel-Bratsch, Switzerland).

### 4.2. shRNA Constructs and Transduction with Lentivirus

ARK1 cells were transduced with lentivirus encoding scrambled sequence (5′-AATTGTACTACACAAAAGTAC-3′, shSCRM) or an AXL shRNA (5′-GATTTGGAGAACACACTGA-3′, shAXL), and selected with puromycin, as previously described [29].

### 4.3. Conditioned Media (CM)

ARK1 cells were counted and plated in equal amounts. The following day after plating, these cells were treated with bevacizumab (250 µg/mL), AVB-500 (2-µM), or both either in 10% or 2% FBS pending the assay. Twenty-four hours after treatment, the conditioned medium was collected and centrifuged at 4000× *g* for 10 min. The supernatant was used for the HUVEC invasion assay, the tube formation assay, and the ELISA assay.

### 4.4. Profiling of Angiogenic Factors Using Cytokine Antibody Array

One million ARK1 shSCRM and shAXL cells were each plated in a 6-cm culture plate. After 24 h, conditioned media were collected and stored at −80 °C. In a separate experiment, one million ARK1 parental cells were plated in 6-cm culture plates for 24 h. Cells were then treated with either vehicle or 1 uM AVB-500. After 24 h of treatment, conditioned media were collected and stored at −80 °C. The relative levels of angiogenic factors secreted by ARK1 shSCRM compared to ARK1 shAXL and by ARK1 vehicle compared to ARK1 + AVB-500 were detected using the RayBio^®^ Human Cytokine Antibody Array C5 (RayBiotech, Inc. Peachtree Corners, GA, USA). Conditioned media from each sample was added to separate membranes. A blocking buffer was added and incubated overnight at 4 °C. After treatment with the washing buffer three consecutive times, the membranes were incubated with biotinylated antibody cocktail overnight at 4 °C. The next morning, the membranes were washed three times, and were incubated with freshly diluted horseradish peroxidase (HRP)–streptavidin overnight at 4 °C. After the membrane was washed, 500 μL of detection buffer mixture was added to each membrane and incubated for 2 minutes at room temperature. After the 2-min incubation, signals were detected using a Bio-Rad developer. The secreted factors were represented by the signal intensity of each spot and were evaluated by subtracting them from the background before being normalized to positive controls using ImageJ software according to the manufacturer’s instructions.

We used the Student’s *t*-test to determine the fold change for each proangiogenic factor. The fold change value represents the level of proangiogenic factors in the conditioned media (CM) from shAXL cells to the conditioned media from shSCRM cells. The fold change of 1 is seen when the proangiogenic factors in the CM from shAXL are the same as the level from the shSCRM cells. Fold change of <1 indicates that the level of proangiogenic factors is lower in the shAXL than shSCRM. A fold change of >1 indicates that the level of proangiogenic factor is higher in the shAXL than shSCRM. Experiments were performed in triplicate and results are expressed as fold change [52].

### 4.5. Measurement of Secreted VEGF

VEGF levels in CM collected as described above were measured by a commercially available ELISA kit (Quantikine^®^ Human VEGF, R&D Systems^®^, Minneapolis, MN, USA). As per the manufacturer’s instructions, the standard curve was generated with VEGF concentrations ranging from 31.2 to 2000 ng/L. All standards and samples were prepared in duplicates and the optical density was measured at 450 nm, with a correction at 550 nm, using an Infinite 200 Pro plate reader (Tecan, Männedorf, Switzerland).

### 4.6. Angiogenesis Invasion Assay

HUVECs were cultured as described above. Biocoat Matrigel Invasion Chambers (Corning) with an 8.0-µm pore polyester membrane were used. Invasion chambers were kept in a −20 °C freezer prior to use. The matrigel in each invasion chamber was primed with 400 µL of ARK1 cancer cell CM for 24 h at 37 °C in a 5% CO_2_ incubator. Fifty thousand HUVECs were mixed in endothelial basal culture medium with no growth supplement and plated on top of the primed matrigel after the CM was aspirated. Subsequently, 750 µL of endothelial culture medium with growth medium supplement was placed in the lower reservoir below the invasion chamber as an attractant. The samples were incubated for 16 h, stained with the HEMA 3 stain set (Thermo Fisher Scientific Waltham, MA, USA), and analyzed. The proportion of cells invading through the matrigel was normalized to the migration through the control inserts [32].

### 4.7. Tube Formation Assay

HUVECs were cultured as described above. Forty-eight well plates (Corning) were used, and Cultrex with a reduced growth factor (RGF) and basement membrane extract (BME) (R&D Systems) were plated in each experimental well. Fifty thousand HUVECs were re-suspended with 1 mL of reduced serum CM from each experimental condition, and 1 × 10^4^ cells/200 µL from each condition were placed atop the RGF BME and incubated at 37 °C 5% CO_2_ for 6 h. Tubes were assessed at 6 h across all conditions with the EVOS^®^ FL cell imaging system (Thermo Fisher Scientific) and analyzed by Wimasis image analysis (https://www.wimasis.com/en/WimTube; accessed on 20 March 2020). Four representative pictures were taken per well, and there were four wells per experimental condition. Multiple parameters were queried, but the total covered area, branch points, total tubes, and total tube length were quantified and analyzed statistically.

### 4.8. Immunohistochemistry

Mouse tumors were placed in 10% formalin for 24 h, washed with Dulbecco’s Phosphate Buffered Saline (Gibco), and placed in 70% ethanol before paraffin embedding. Five-micrometer sections were cut and mounted on glass slides. Slides were deparaffinized in 60 °C incubator and hydrated in Xylene. They were subsequently diluted in ethanol and water before being placed in Sodium Citrate at pH of 6 for antigen retrieval. Slides were probed with anti-CD31 rabbit polyclonal antibody (1:50, ab28364, Abcam) at 4 °C for 16 h. The slides were subsequently washed, conjugated with biotinylated anti-rabbit IgG (H + L) (Vector Laboratories, #BA-1000) secondary antibody, and incubated with HRP. Antibody complexes were detected with 3,3′-diaminobenzidine and counter stained with Mayer’s hematoxylin. Stained vessels were imaged with the Nikon Eclipse E800 and images were exported into ImageJ for counting and quantification [53]. The mean vessel density was calculated by averaging vessel counts from 4 random fields per slide, which is similar to what has been previously reported in the literature [54].

### 4.9. Western Blot Analysis

Protein lysates were harvested from cultured cells using 9 mol/L of urea and 0.075 mol/L of Tris buffer (pH 7.6), and quantified using the Bradford assay (BioRad). Seventy-five micrograms of protein was subjected to reducing SDS/PAGE by standard methods then transferred onto a 0.22-µm nitrocellulose membrane (BioRad). Western blots were probed with primary antibodies against AXL (1:1000, #8661) phospho-AXL Tyr702 (1:500, #5724), AKT (1:1000, #9272), and phospho-AKT Ser473 (1:500, #9271) from Cell Signaling Technology and GAS6 (1:1000, #AF885) from R&D Systems and corresponding secondary peroxidase-conjugated anti-rabbit IgG (H + L) (Jackson ImmunoResearch, # 711-035-152), anti-mouse IgG (H + L) (Jackson ImmunoResearch, # 711-035-150), or anti-goat IgG (H + L) (Jackson ImmunoResearch, # 711-035-180) (Appendix A). To confirm equal protein loading, blots were probed with antibodies specific for β-actin (1:3000, #A1978) from Sigma-Aldrich. Quantification was completed with Image Lab Software (BioRad) using box volume tool as per the manufacturer’s suggestions (https://www.bio-rad.com/en-us/applications-technologies/image-analysis-quantitation-for-western-blotting?ID=PQEERM9V5F6X accessed on 4 August 2020). pAKT and total AKT protein signal intensities were first quantified with local background subtraction. pAKT to total AKT ratios were then calculated for each sample and normalized to a pAKT/AKT ratio in the control group by dividing values of pAKT/AKT ratios in all treatment groups by a pAKT/AKT ratio in the control group.

### 4.10. Orthotopic Model of USC

All animal experiments were carried out in accordance with the guidelines of the American Association for Accreditation for Laboratory Animal Care and the U.S. Public Health Service Policy on Humane Care and Use of Laboratory Animals and were approved by the Washington University Institutional Animal Care and Use Committee in accordance with the Animal Welfare Act, the Guide for the Care and Use of Laboratory Animals, and NIH guidelines.

Tumor models were established by injecting 1 × 10^7^ ARK1 cells intraperitoneally (IP) into 6- to 8-week-old female (NOD) SCID mice (Jackson Laboratories Bar Harbor, ME, USA). The mice were continually monitored throughout the course for adverse events until sacrifice. Beginning six days after tumor cell injections, mice were intraperitoneally injected every 3 days for 21 days with 5 mg/kg of bevacizumab (Washington University Pharmacy), 30 mg/kg of AVB-500 (Aravive, Inc. Houston, TX, USA), or Dulbecco’s Phosphate Buffered Saline (Gibco Inc. Billings, MT, USA) supplemented with magnesium chloride (MgCl_2_) and calcium chloride (CaCl_2_). Mice were sacrificed on day 27.

A pilot study was conducted for the ARK4 IP orthotopic tumor model. Female 6- to 8-week-old (NOD) SCID mice (*n* = 5) were injected with 1 × 10^7^ ARK4 cells. The mice were sacrificed at weekly intervals, revealing that visible tumor burden was evident on day 35. Subsequently, 24 mice were injected with the same number of ARK4 cells. After seven days, treatment began every three days, as described above. The mice were sacrificed on day 35. One mouse from the bevacizumab only group died two days before sacrifice (presumably due to tumor burden) and was excluded from statistical analysis.

### 4.11. Statistical Analysis

GraphPad Prism software was used for statistical analyses. Two-tailed unpaired Student’s t-tests were performed with ANOVA where appropriate, to analyze differences between groups. *p* < 0.5 was considered statistically significant.

## 5. Conclusions

In conclusion, our preclinical data suggest a novel and effective option for treatment of USC with a therapeutic AXL inhibitor, AVB-500, in conjunction with bevacizumab. To our knowledge, it is the first study that has demonstrated the efficacy of AXL inhibition and an antiangiogenic in USC, thus warranting further study.

## Figures and Tables

**Figure 1 cancers-13-05877-f001:**
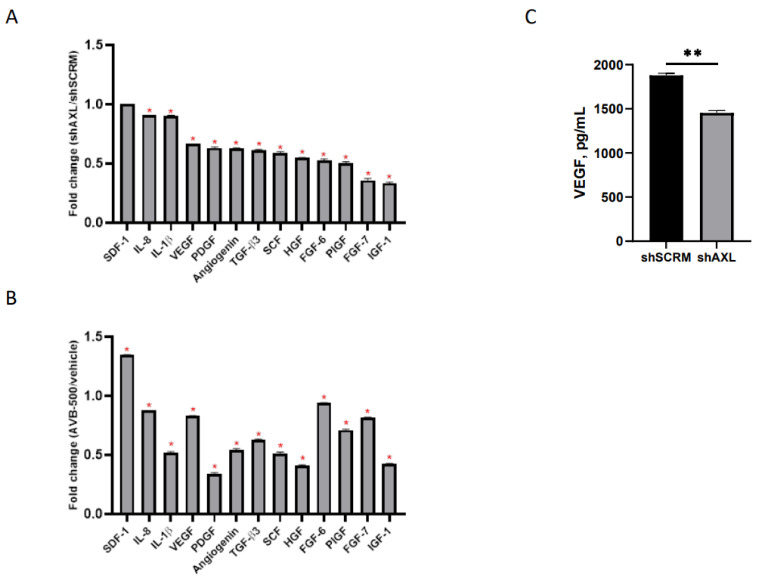
Regulation of angiogenic factors in ARK1 cells by AXL. (**A**) Conditioned media from ARK1shSCRM or ARK1shAXL cells and (**B**) Conditioned media from ARK1 cells treated with AVB-500 vs. vehicle were collected and subjected to cytokine profiling using RayBiotech cytokine array 5. The cytokine array image was analyzed using ImageJ and quantification of the signal intensity of indicated factors were plotted. Fold change was measured between each group and significant change in pro-angiogenic factors is noted by red asterisk. Fold change of 1 equals no change in shSCRMand shAXL(SDF-1), <1 indicates proangiogenic factors lower in shAXL. * *p* < 0.001. (**C**) Conditioned media from ARK1shSCRM or ARK1shAXL cells were collected and analyzed with the VEGF ELISA assay. Data are presented as absolute protein values (pg/mL). ** *p* < 0.01.

**Figure 2 cancers-13-05877-f002:**
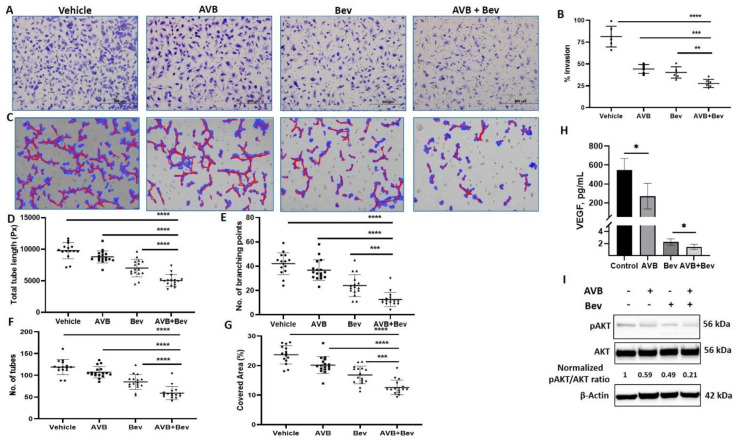
Inhibition of AXL and VEGF-A reduces endothelial cell invasion, tube formation, and protein expression. (**A**) Representative images of HUVEC invasion in each treatment condition. Scale bar, 500 µm. (**B**) Quantitation of invasion in the indicated conditions. (**C**) Representative Wimasis images of tube assays. Red lines indicate tubes and blue indicates covered area of the tubes. (**D**) The total tube length in image pixels (Px). (**E**) The number of branching points, and (**F**) the number of tubes. (**G**) The percentage of covered area. (**H**) VEGF concentration in conditioned media of ARK1 cells treated with AVB and bevacizumab as single agents or in combination. Bar represents means ± SD. (**I**) Western blot of HUVEC cells treated with AVB and bevacizumab as single agents or in combination. Equal quantities of whole cell lysates from each sample were subjected to immunoblotting with specific antibodies, as indicated. β-Actin used as loading control. Normalized pAKT/AKT ratios are indicated. Significant differences between groups: * *p* < 0.05, ** *p* < 0.01, *** *p* < 0.001, **** *p* < 0.0001.

**Figure 3 cancers-13-05877-f003:**
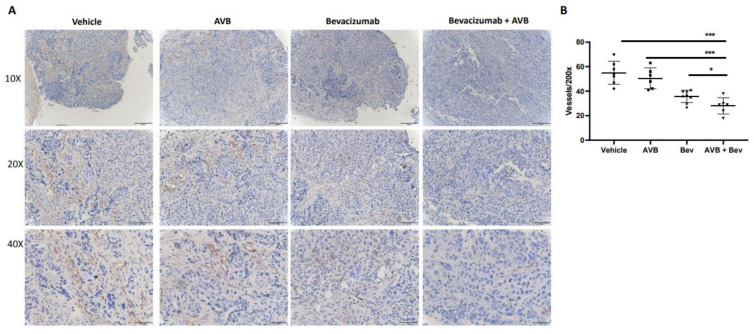
IHC CD31 staining of in vivo tumors shows decreased microvessel density in the AVB + Bev group. (**A**) Representative images of CD31 staining in each condition at different magnifications-10×, 20×, and 40×. Scale bar, 50 µm, 100 µm, 200 µm, respectively. (**B**) The dot plot shows a decrease in CD31 vessels at 200× magnification. * *p* < 0.05, *** *p* < 0.001.

**Figure 4 cancers-13-05877-f004:**
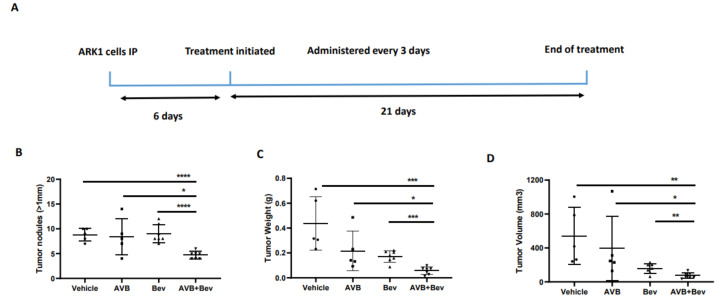
Inhibition of AXL and VEGF-A in vivo, demonstrates decreased tumor burden in a USC cell line, ARK1. (**A**) Schematic showing the time period of the ARK1 in vivo experiment. Dot plots showing decreases in (**B**) tumor weight (**C**) tumor volume and (**D**) tumor nodules measured to be over 1mm in the AVB + Bev group. The bars indicate comparisons between each treatment condition and the AVB + Bev combination group. * *p* < 0.05, ** *p* < 0.01, *** *p* < 0.001, **** *p* < 0.0001.

**Figure 5 cancers-13-05877-f005:**
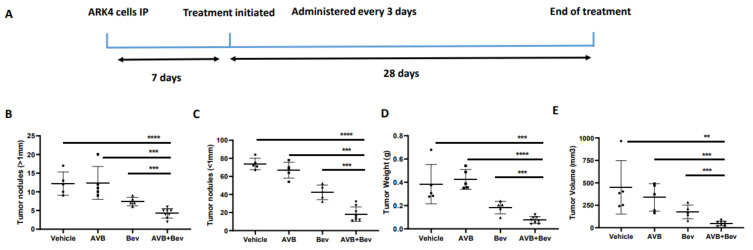
Inhibition of AXL and VEGF-A in vivo demonstrates decreased tumor burden in an alternate USC cell line, ARK4. (**A**) Schematic showing the time period of the ARK 4 in vivo experiment. Dot plots showing decreases in (**B**) tumor weight, (**C**) tumor volume, (**D**) tumor nodules measured to be over 1 mm, and (**E**) tumor nodules measured to be under 1 mm in the AVB + Bev group. The bars indicate comparisons between each treatment condition and the AVB + Bev combination group. ** *p* < 0.01, *** *p* < 0.001, **** *p* < 0.0001. (**F**) Representative images from the in vivo experiment. Red circles indicate tumor burden visualized intraperitoneally. AVB + Bev mouse has tumor limited to the pelvis with no upper abdominal disease visualized.

**Figure 6 cancers-13-05877-f006:**
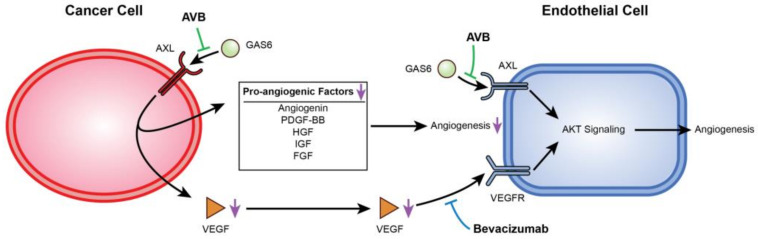
Mechanistic display of how inhibition of VEGF-A and AXL negatively regulate angiogenesis. This figure displays the mechanism between USC cells and HUVECs when AXL and VEGF-A are inhibited. AVB-500 blocks GAS6 from binding to AXL downregulating angiogenic factors. Simultaneously, both AVB-500 and bevacizumab inhibit activation of AXL and the VEGF receptor on the HUVECs.

**Table 1 cancers-13-05877-t001:** Summarizes percentage of mice with ascites present, percentage of mice with diaphragmatic metastases, and estimated volume of ascites in each treatment condition at the time of sacrifice.

	Vehicle (*n* = 5)	AVB (*n* = 5)	Bev (*n* = 5)	AVB + Bev (*n* = 8)
Diaphragmatic metastases (%)	100	100	60	12.5 ***
Ascites (%)	100	100	40	12.5 ***
Mean volume ascites (µL)	620	340	24	0.625 ^^^

*** Indicates *p* < 0.001 when comparing AVB + Bev to AVB or vehicle; ^^^ Indicates *p* < 0.01 when comparing AVB + Bev to vehicle and *p* < 0.05 when comparing AVB + Bev to AVB.

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
