# Peer review of "Inhibition of AXL and VEGF-A Has Improved Therapeutic Efficacy in Uterine Serous Cancer"

_cancers, 2021, doi:10.3390/cancers13235877_

Round 1

Reviewer 1 Report

In this work, the authors showed that through the inhibition of receptor tyrosine kinase AXL with AVB-500 in combination with bevacizumab, uterine serous cancer decreased angiogenic potential, tumor burden and educed morphogenesis and migration in vitro.

The argument of the presented study is very interesting given the high interest in the endometrial cancer incidence, that stimated 66,570 cases and 12,940 deaths predicted in the United States in 2021.

Specific comments for revision:

Short 2-3 sentences as introduction should be written for the beginning of subsections “Inhibition of AXL and VEGF-A shows relative decrease in VEGF” and “IHC CD31 staining of mouse tumors shows decreased vessel density

Increase the font size on the figures 1, 2, 3

Impossible to see the “Red lines indicate tubes and blue indicates covered area of the tubes” on Fig 2C – authors should increase the quality of the photos

Molecular weight for western blot photos and densitometric analysis should be added to fig 2I

Materials and methods – add code for secondary antibody for western blotting

The authors should redesign the graphical abstract in an appropriate professional programme.

Author Response

Response to reviewers attached in a word document.

Reviewer 2 Report

The authors present an interesting article on the therapeutic targeting of an under studied disease in uterine serous cancer.  Dual targeting of AXL and VEGF-A in combination seems like an attractive approach and adds merit to the current clinical trial that is ongoing with these cell line and mouse model experiments. Overall the experiments are sound and well thought out but I have a number of small concerns with the manuscript in its current form. 

Figure 1: Fold changes seem quite modest to represent such significant p values and have no real error bars? Can I ask the authors to plot each replicate as an individual data point in this data set. Also, what statistical test was performed on this data and what are each of the values compared to? Currently this is not clear. Figure 1A looks like everything is relative to SDF-1 and the other markers decrease accordingly. Data could be presented in 1A and B more clearly and the analysis of the array potentially revisited. 

Also Figure 1C is not referenced in the figure legend. Absolute protein values (pg/ml) should be used and again n numbers and data points clearly identifiable on the bar charts. 

Suggest revisiting figure legend in figure 2 as mention of figure 2H is currently incorrect and a repeat of Figure 2I. Also WB is not clear that the decrease is significant. How many reps were done? This is not clear from the figure legend. Suggest using an alternative image. 

If figure 2 (H) is an ELISA, again, use absolute values rather that % to control. 

Can I ask the authors to include any data regarding ARK-4 cells treated with AUB-500 in the overall manuscript as these cell lines are the low to null HER2 expression model. Experiments should be reflective of both HER2+ and - expression. 

Check timeline of treatment figure in Figure 5A as it states its ARK1 cells IP not ARK4 as suggested by the figure legend.

The mouse model experiments are convincing however, but I suggest the authors revisit some of their cell line data before publishing. 

Author Response

Response to reviewers uploaded in a word document. 

Round 2

Reviewer 2 Report

I thank the authors for their reply and I have no further comments.